# Exploring the relationship between pancreatic fat and insulin secretion in overweight or obese women without type 2 diabetes mellitus: A preliminary investigation of the TOFI_Asia cohort

Ivana R. Sequeira[1,2]*, Wilson Yip[1,2], Louise W. Lu[1,2], Yannan Jiang[3], Rinki Murphy[2,4,5,6], Lindsay D. Plank[7], Garth J. S. Cooper[8,9,10], Carl N. Peters[4,11], Benjamin S. Aribsala[12,13], Kieren G. Hollingsworth[12], Sally D. Poppitt[1,2,4,14]

1 Human Nutrition Unit, School of Biological Sciences, Faculty of Science, University of Auckland, Auckland, New Zealand, 2 High-Value Nutrition National Science Challenge, Auckland, New Zealand, 3 Department of Statistics, Faculty of Science, University of Auckland, Auckland, New Zealand, 4 Department of Medicine, Faculty of Medical and Health Sciences, University of Auckland, Auckland, New Zealand, 5 Auckland District Health Board, Auckland, New Zealand, 6 Maurice Wilkins Centre for Molecular Biodiscovery, University of Auckland, Auckland, New Zealand, 7 Department of Surgery, Faculty of Medical and Health Sciences, University of Auckland, Auckland, New Zealand, 8 Division of Cardiovascular Sciences, Centre for Advanced Discovery and Experimental Therapeutics (CADET), Faculty of Biology, Medicine and Health, University of Manchester, Manchester, United Kingdom, 9 School of Biological Sciences, Faculty of Science, University of Auckland, Auckland, New Zealand, 10 Division of Medical Sciences, Department of Pharmacology, University of Oxford, Oxford, United Kingdom, 11 Waitemata District Health Board, Auckland, New Zealand, 12 Newcastle Magnetic Resonance Centre, Translational and Clinical Research Institute, Faculty of Medical Science, Newcastle University, Newcastle Upon Tyne, United Kingdom, 13 Department of Computer Science, Faculty of Science, Lagos State University, Lagos, Nigeria, 14 Riddet Centre of Research Excellence (CoRE) for Food and Nutrition, Palmerston North, New Zealand

* i.sequeira@auckland.ac.nz

**Data Availability Statement:** De-identified data will be shared and made available upon reasonable

## Abstract

### Objective

While there is an emerging role of pancreatic fat in the aetiology of type 2 diabetes mellitus (T2DM), its impact on the associated decrease in insulin secretion remains controversial. We aimed to determine whether pancreatic fat negatively affects β-cell function and insulin secretion in women with overweight or obesity but without T2DM.

### Methods

20 women, with normo- or dysglycaemia based on fasting plasma glucose levels, and low (< 4.5%) *vs* high ($\geq$ 4.5%) magnetic resonance (MR) quantified pancreatic fat, completed a 1-hr intravenous glucose tolerance test (ivGTT) which included two consecutive 30-min square-wave steps of hyperglycaemia generated by using 25% dextrose. Plasma glucose, insulin and C-peptide were measured, and insulin secretion rate (ISR) calculated using regularisation deconvolution method from C-peptide kinetics. Repeated measures linear mixed models, adjusted for ethnicity and baseline analyte concentrations, were used to compare changes during the ivGTT between high and low percentage pancreatic fat (PPF) groups.

request to the authors and Ethics committee (hdecs@health.govt.nz) subject to an approved proposal and data access agreement in accordance with Ethical approval received from the Health and Disability Ethics Committee (16/STH/23).

**Funding:** This study was funded by the New Zealand National Science Challenge High Value Nutrition Program, Ministry for Business, Innovation and Employment (MBIE, grant no. 3710040). Funding acquisition: SDP The funders had no role in study design, data collection and analysis, decision to publish, or preparation of the manuscript.

**Competing interests:** The authors have declared that no competing interests exist.

**Abbreviations:** AAT, abdominal adipose tissue; BMI, Body-mass index; DAG, diacylglycerol; DBP, diastolic blood pressure; DXA, dual energy X-ray absorptiometry; FGIR, fasting glucose to insulin ratio; FPG, fasting plasma glucose; FFA, free fatty acid; iAUC, incremental area under the curve; IFG, impaired fasting glucose; IR, insulin resistance; IS, insulin sensitivity; ISR, insulin secretion rate; ivGTT, intravenous glucose tolerance test; LOD, limit of detection; MRI/S, magnetic resonance imaging and spectroscopy; OGTT, oral glucose tolerance test; PPF, percentage pancreatic fat; SAT, subcutaneous adipose tissue; SBP, systolic blood pressure; TOFI, thin-on-the-outside-fat-on-the-inside; TBF, total body fat; TAG, triglyceride; VAT, visceral adipose tissue.

## Results

No ethnic differences in anthropomorphic variables, body composition, visceral adipose tissue (MR-VAT) or PPF were measured and hence data were combined. Nine women (47%) were identified as having high PPF values. PPF was significantly associated with baseline C-peptide (p = 0.04) and ISR (p = 0.04) in all. During the 1-hr ivGTT, plasma glucose (p<0.0001), insulin (p<0.0001) and ISR (p = 0.02) increased significantly from baseline in both high and low PPF groups but did not differ between the two groups at any given time during the test (PPF x time, p > 0.05). Notably, the incremental areas under the curves for both first and second phase ISR were 0.04 units lower in the high than low PPF groups, but this was not significant (p > 0.05).

## Conclusion

In women with overweight or obesity but without T2DM, PPF did not modify β-cell function as determined by ivGTT-assessed ISR. However, the salient feature in biphasic insulin secretion in those with ≥4.5% PPF may be of clinical importance, particularly in early stages of dysglycaemia may warrant further investigation.

## Introduction

Increased visceral adipose tissue (VAT) is said to mediate insulin resistance (IR) and lipotoxicity with resultant lipid accumulation in non-adipose ectopic sites [1], and may well contribute to ethnic-specific susceptibility to type 2 diabetes mellitus (T2DM) [2, 3]. Several clinical studies have reported that pancreatic fat content negatively correlates with insulin secretion [4–6], using magnetic resonance imaging and spectroscopy (MRI/S) and oral glucose tolerance tests (OGTT) in populations with and without diabetes, while others have not found such a correlation [7–9]. Pre-clinical models have established that fat accumulation in islet cells [10, 11] decreases mitochondrial beta-oxidation, thereby increasing intracellular triglycerides (TAG) [12] and dampening glucose-mediated insulin secretion [13]. However, the impact of increased pancreatic fat on the reduced insulin secretion component of impaired glucose homeostasis remains controversial. Lack of reported correlation between indices of insulin secretion and pancreatic fat may be attributable to an individual cellular threshold [14] for toxic intermediates rather than a simple dose response relationship within the pancreas. It is hypothesised that toxic intermediates from pancreatic TAG, including ceramides and diacylglycerol (DAG) rather than the stored TAG itself, may be responsible for interfering with pancreatic β-cell insulin secretion.

Whilst lowering of liver fat below 5.6%, corresponding to ~15% histological liver fat [15], has been associated with normalisation of blood glucose, no predetermined pancreatic fat cut-off is yet recognised. A recommendation for a normal upper limit of 6.2% pancreatic fat has been reported in a meta-analysis from 9 studies in 1209 healthy individuals [16]; however, this was based on MR studies utilising different methods and quantitative approaches, potentially limiting reproducibility. Notably, fat infiltration occurs within exocrine, interlobular, and acinar cells, and varies among ethnic groups [17, 18]; however, the association with endocrine β-cell function is yet to be determined. Expectedly, a reported 2- to 3-fold increase in insulin secretion is observed in individuals with obesity [19] and prediabetes [20–22] due to β-cell compensation, which is shown to eventually lead to β-cell damage and reduction in β-cell

mass/volume. As this process progresses, reduced β-cell mass and function can no longer compensate for IR, causing hyperglycaemia to further damage β-cell cells due to glucotoxicity and contributes to T2DM progression.

Szczepaniak *et al.* [6] demonstrated that pancreatic fat content and the rate of insulin secretion differed between ethnicities, i.e. Caucasian and Hispanics with normal glucose tolerance. In line with these findings, a recent systematic review and meta-analysis showed that the insulin secretory response in Asians, comprising Chinese, was lower than Caucasians for the same level of insulin sensitivity, although body fat or pancreatic fat was not reported [23]. However, causal mechanisms and pathways that link pancreatic fat deposition and β-cell insulin secretion remain to be elucidated. The rate of insulin secretion derived from an intravenous glucose tolerance test (ivGTT) is considered to be a "gold-standard" measurement to assess β-cell function. The paucity of published studies [14] using the ivGTT and standardised MR methodology for quantification of intra-pancreatic fat therefore led us to undertake this investigation in a sub-set of women from the thin-on-the-outside-fat-on-the-inside (TOFI)_Asia cross-sectional study. We hypothesised that a stepped ivGTT would provide a robust method to determine whether β-cell response may be compromised in women with pancreatic fat above an arbitrary conservative 4.5% cut off, alongside a reduction in first-phase insulin release. Importantly, as illustrated by Bergman [24] and Kahn [25], given that islet cells adjust their function to compensate for IR, prevailing insulin sensitivity (IS), glycaemic status on the morning of the test and ethnicity were included in determining the relationship between pancreatic fat and β-cell function.

## Materials and methods

The ivGTT study in the TOFI_Asia MR cohort [26] was conducted at the Human Nutrition Unit (HNU), University of Auckland with ethical approval obtained from the Health and Disability Ethics Committee (16/STH/23) and was registered with the Australian New Zealand Clinical Trials Registry (ACTRN12616000362493).

### Study population

Of the 202 women who were enrolled in the TOFI_Asia cross-sectional study, a subset of 68 women (34 Chinese and 34 Caucasian) successfully joined the TOFI_Asia MR cohort and underwent abdominal MRI scans for assessment of pancreatic fat. Of these, 22 women (9 Chinese and 13 Caucasian) provided written informed consent and were enrolled into the current ivGTT cohort, for determination of pancreatic β-cell function as a measurement of insulin secretion (Fig 1). Informal power calculations showed a minimum of 10 individuals would be required to detect a physiologically significant increase in first phase insulin response. Inclusion criteria was based on the TOFI_Asia MR study [26] and as such the ivGTT cohort were 20–70 years of age, body mass index (BMI) 20–35 kg/m$^2$, normoglycaemic (fasting plasma glucose < 5.6 mmol/L) or impaired fasting glucose (IFG; 5.6–6.9 mmol/L) [27] and MRI-assessed percentage pancreatic fat (PPF) classified as < 4.5% or ≥ 4.5%. PPF could not be analysed from scans obtained from 2 women due to motion related artefact, and hence 20 women (9 Chinese and 11 Caucasian) had an assessment of pancreatic β-cell function using ivGTT and were included in the current analyses (Fig 1).

### Study design and protocols

All women enrolled in the TOFI_Asia MR cohort attended the HNU clinic following an overnight fast; body weight, height, waist and hip circumferences, and blood pressure (BP) were recorded, and a fasted blood sample for determination of HbA$_{1c}$ collected, stored at -80˚C

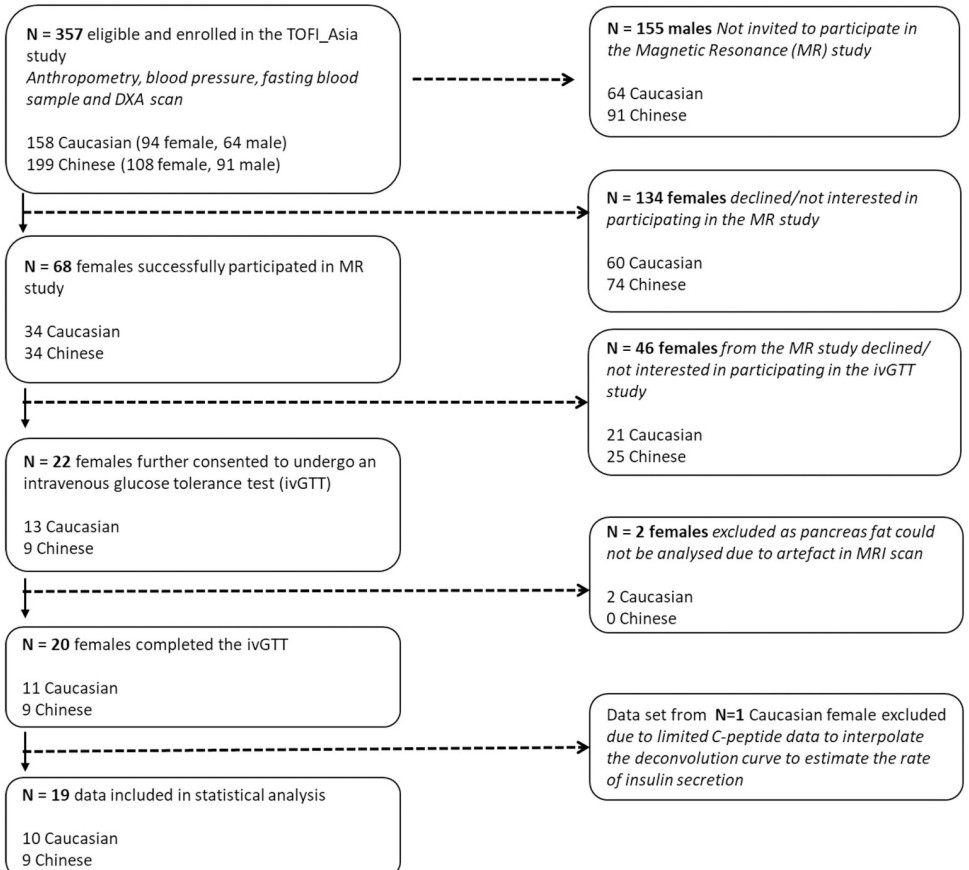

**Fig 1. Flow chart illustrating enrolment and eligibility of female participants from the TOFI_Asia study for the ivGTT study.** *Nonstandard abbreviations*: DXA, dual-energy X-ray absorptiometry; MR, magnetic resonance.

pending batch analysis on completion of the study using capillary electrophoresis (Cap2FP, IDF, France). Total and abdominal adiposity were assessed using dual-energy X-ray absorptiometry (DXA) (iDXA, GE Healthcare, WI, USA) at the Body Composition Laboratory, University of Auckland. Detailed abdominal MRI/S scans were conducted in the fasted state, within 1 week of the clinic visit, on a 3T Magnetom Skyra scanner, VE 11A (Siemens, BY, Germany) at the Centre for Advanced MRI (CAMRI), University of Auckland.

**Anthropometry and BP measurement.** Height (cm, Seca 222, Hamburg, Germany) and weight (kg, Mettler Toledo Spider, Greifensee, Switzerland) were recorded wearing light clothing and without shoes. Waist circumference was measured midway between the inferior margin of the lower rib and the iliac crest and hip circumference at the widest point over the greater trochanter. Blood pressure was recorded using a digital Critikon Dinamap sphygmomanometer (GE Healthcare, Shanghai, China) with participant seated and rested.

**Assessment of body composition by DXA.** Scans were conducted according to the manufacturer's standard protocols. Abdominal adipose tissue (AAT) and lean (fat-free soft) tissue masses were determined in a region of interest (ROI), set automatically with a caudal limit at the top of the iliac crest, and height set to 20% of the distance from this limit to the base of the skull. Percent total body fat (%TBF) was calculated as [100 x TBF mass/(total body lean mass + TBF mass)] and %AAT as [100 x AAT mass/(abdominal lean mass + AAT mass)].

**Quantification of abdominal, pancreas and liver fat by MRI/S.** A 3D dual gradient-echo sequence (VIBE) was used to acquire separate fat and water signals using the 2-point Dixon technique: repetition time/echo times/flip angle = 3.85 ms/1.27, 2.50 ms/9°. Three blocks of forty 5-mm axial slices, field of view (FOV) 500 x 400 mm, matrix 320 x 256, were acquired to cover the abdominal cavity using partial Fourier and parallel imaging with total acceleration factor of 3.1. Thereafter, fourteen pancreas 5-mm axial slices were acquired at repetition time/echo times/flip angle/signal averages = 5.82 ms/2.46, 3.69 ms/9°/1 with FOV 500 x 400 mm, matrix 512 x 410, using partial Fourier and parallel imaging with total acceleration factor of 2.8. Localizer images were obtained in the transverse, coronal and sagittal planes and a voxel (2 x 2 x 2 cm$^3$) placed in the right liver lobe. MRS of the voxel was acquired using the stimulated-echo acquisition mode sequence with respiratory triggering; echo time: 20 msec, repetition time: 3000 msec and mixing time: 33 msec, 1024 data points collected with 50 averages.

Following image acquisition, abdominal VAT and subcutaneous adipose tissue (SAT), at the L5 intervertebral disc space, was determined by segmentation using ImageJ [28] and pancreatic fat quantified by MR-opsy [29]. There is no global comparable cut off for PPF, particularly since the pancreas imaging in this study has T1 weighting and will overestimate fat fraction by a factor of approximately 1.4 [30]. A PPF value of 4.5% was used to categorise high ($\geq$ 4.5%) or low ($<$ 4.5%) pancreatic fat. Pancreas volume (cm$^3$) and pancreas volume index (PVI) were calculated as previously described [31]. Liver fat was calculated from area under the curve of water and fat peaks from non-water-suppressed spectra in SIVIC software [32].

**Assessment of pancreatic β-cell function.** Following MR scanning, women who provided additional written informed consent were scheduled for their ivGTT within 7 days. A stepped insulin secretion test was performed as previously described [14, 33]. In fasted participants, two 18G venous cannulae were inserted into the antecubital vein for infusion and the contralateral wrist vein for arterialised blood sampling. The hand was kept warm during the entire test to maintain arterialisation of venous blood. Two consecutive, 30-min square-wave steps of hyperglycaemia (2.8 and 5.6 mmol/L above baseline) were achieved using a bolus dose of 25% dextrose (Baxter Healthcare, NSW, Australia). The priming bolus dose for each step was calculated as 0.378 g/kg body weight of 25% dextrose and administered over one min to avoid thrombophlebitis. Post-administration of the bolus dose, plasma glucose concentrations were maintained at the desired plateau with a variable 25% dextrose infusion (Carefusion, Alaris®, BBRK, UK). Blood samples for determination of plasma glucose, insulin and C-peptide were obtained every 2 min for the first 10 min and every 5 min over the next 20 min of each step. Two Caucasian women did not successfully complete the second 30-min of the test, due to the inability to obtain arterialised venous blood. We were unable to obtain sufficient blood samples for analysis and estimation of insulin secretion rate (ISR) from one Caucasian woman, who was therefore excluded from the statistical analysis. Therefore, final data presented are for 19 women (9 Chinese and 10 Caucasian) (Fig 1).

Plasma glucose (limit of detection [LOD]: 0.11 mmol/L) was measured by the hexokinase method (COBAS c311, Roche Diagnostics, BW, Germany), plasma insulin (LOD: 87 pg/mL) and C-peptide (LOD: 9·5 pg/mL) were analysed using the multiplex immunoassay (MILLI-PLEX®MAP Human Metabolic Hormone Magnetic Bead Panel, Merck, HE, Germany). The fasting glucose to insulin (FGIR) and fasting C-peptide to glucose ratios [34] were calculated as indices of IS and β-cell function respectively. ISR was determined over the test period using a computerised program implementing a regularisation deconvolution method [35] based on C-peptide kinetics [36]. First and second phase Δ peak ISR were calculated by subtracting peak values from baseline and 30 min, respectively.

## Statistical analysis

Data are presented as mean ± SD for all continuous variables in descriptive summaries. Group comparisons were performed using *two-sample* tests appropriate to the distribution of variables. Pearson correlations and linear regression analysis were used to determine relationships between metabolic risk factors at baseline. Missing data points for blood analytes were imputed and assigned the mean value calculated for that participant over the 60-min test. First-phase insulin response was calculated as the incremental area under the curve (iAUC) of the ISR peak at 10 min, and second-phase insulin response was calculated as iAUC at 30–40 min. Changes from baseline (Δ) in glucose, insulin, C-peptide and ISR over the 60-min test were evaluated using repeated measures linear mixed models, adjusted for ethnicity and the average baseline (-10 min and 0 min) concentration, in women identified with PPF above and below our defined high/low PPF cut-off. The interaction effect between the two PPF groups and time points was tested in the model. Statistical tests were two-sided at 5% significance level. All statistical analyses were conducted using SAS® Version 9.4 (SAS Institute Inc., Cary, 245 NC, USA).

## Results

### Participant characteristics

The participants included in the current analysis (mean age 47.3 ± 12.2 y; BMI 27.0 ± 3.7 kg/m$^2$) comprised 47% Chinese (n = 9; mean age 46.8 ± 10.1 y; BMI 26.2 ± 3.3 kg/m$^2$) and 53% Caucasian women (n = 10; mean age 47.7 ± 14.4 y; BMI 27.7 ± 4.1 kg/m$^2$) (Table 1). 63% (n = 12) women were normoglycaemic (mean fasting plasma glucose; FPG 5.5 ± 0.8 mmol/L), while 37% (n = 7) were identified as IFG (mean FPG 6.2 ± 0.8 mmol/L) with an equal division between the ethnic groups (n = 4 Chinese; mean FPG 6.4 ± 1.0 mmol/L, n = 3 Caucasian; mean FPG 6.0 ± 0.6 mmol/L) (Table 1). 58% of the women (n = 11; 5 Chinese, 6 Caucasian) were overweight, 26% (n = 5; 3 Chinese, 2 Caucasian) had obesity and 16% (n = 3; 1 Chinese, 2 Caucasian) had normal weights.

Chinese women were of similar mean age and BMI to their Caucasian counterparts, with mean body composition as assessed by DXA and MRI also similar between the two ethnic groups (Table 1). Exceptions were MR-determined abdominal subcutaneous adipose tissue (MR-SAT cm$^2$) which was significantly lower and Hb$_{A1c}$ significantly higher in Chinese than Caucasian (both p = 0.04) (Table 1).

The range of PPF was 1.9–7.1% in the full cohort. While mean PPF was 0.3% below our defined high/low PPF cut-off (Table 1), 9 women (n = 5 Chinese, n = 4 Caucasian) were identified with high PPF ≥ 4.5% (mean FPG 5.6 ± 0.9 mmol/L, HbA$_{1c}$ 34.6 ± 2.9 mmol/mol) and 10 women (n = 4 Chinese, n = 6 Caucasian) with low PPF < 4.5% (mean FPG 5.4 ± 0.6 mmol/L, HbA$_{1c}$ 35.1 ± 5.5 mmol/mol) (Table 2).

Anthropometric and body composition parameters for these high and low PPF groups are presented in Table 2. Despite similar mean body weight and BMI between groups, DXA-TBF and DXA-AAT were both numerically higher in women with high PPF, although not statistically significant in this small cohort. Notably, the high group PPF had significantly greater MR-determined VAT (MR-VAT, p = 0.05) and significantly lower PVI (p = 0.03). Also, % liver fat was two-fold greater (10.7 ± 8.6% *vs* 5.0 ± 4.9%, p = 0.09) in the high PPF group.

### Relationships between baseline circulating biomarkers, ISR and indices of insulin sensitivity with PPF

Despite no significant relationship with baseline FPG or fasting insulin, PPF was significantly and positively associated with both fasting C-peptide (p = 0.04) and ISR (p = 0.04) (Fig 2C and 2D) in

**Table 1. Summary characteristics of baseline anthropometry and body composition variables, determined by dual-energy X-ray absorptiometry and magnetic resonance imaging, in Chinese and Caucasian women enrolled in the ivGTT study.**

| | All (N = 19) | Chinese (N = 9) | Caucasian (N = 10) | *p value* |
|---|---|---|---|---|
| Age (y) | 47.3 ± 12.2 | 46.8 ± 10.1 | 47.7 ± 14.4 | 0.88 |
| *Anthropometry* | | | | |
| Body weight (kg) | 72.5 ± 13.6 | 66.5 ± 7.4 | 77.9 ± 16.0 | 0.07 |
| Height (m) | 1.64 ± 0.08 | 1.59 ± 0.05 | 1.67 ± 0.09 | 0.04 |
| Body mass index, BMI (kg/m$^2$) | 27.0 ± 3.7 | 26.2 ± 3.3 | 27.7 ± 4.1 | 0.39 |
| Waist circumference (cm) | 88.6 ± 11.1 | 87.0 ± 8.6 | 90.1 ± 13.3 | 0.56 |
| Hip circumference (cm) | 100.7 ± 10.0 | 96.8 ± 7.5 | 104.2 ± 11.0 | 0.11 |
| Systolic blood pressure, SBP (mmHg) | 124 ± 24 | 126 ± 28 | 122 ± 20 | 0.70 |
| Diastolic blood pressure, DBP (mmHg) | 67 ± 14 | 68 ± 19 | 64 ± 7 | 0.51 |
| *Body composition—DXA* | | | | |
| Total body fat, DXA-TBF (kg) | 28.5 ± 9.2 | 25.0 ± 4.3 | 31.5 ± 11.5 | 0.13 |
| Total body fat, DXA-TBF (%) | 40.0 ± 6.2 | 38.8 ± 3.4 | 41.1 ± 8.1 | 0.45 |
| Abdominal adipose tissue, DXA-AAT (kg) | 2.4 ± 0.9 | 2.2 ± 0.5 | 2.6 ± 1.2 | 0.35 |
| Abdominal adipose tissue, DXA-AAT (%) | 45.0 ± 8.2 | 44.6 ± 4.9 | 45.4 ± 10.7 | 0.84 |
| *Body composition–MRI/S* | | | | |
| Visceral adipose tissue, MR-VAT (cm$^2$) | 69.5 ± 24.1 | 72.3 ± 22.4 | 67.1 ± 26.5 | 0.65 |
| Subcutaneous adipose tissue, MR-SAT (cm$^2$) | 165.8 ± 61.5 | 136.2 ± 20.2 | 192.4 ± 74.4 | 0.04 |
| Liver fat (%) | 7.7 ± 7.3 | 7.7 ± 5.3 | 7.7 ± 9.0 | 0.43 |
| Pancreatic fat (%) | 4.2 ± 1.7 | 4.4 ± 1.7 | 4.0 ± 1.8 | 0.57 |
| Pancreas volume cm$^3$ | 74.3 ± 19.4 | 73.9 ± 21.1 | 74.6 ± 18.8 | 0.94 |
| Pancreas volume index, PVI | 42.0 ± 11.2 | 43.9 ± 13.0 | 40.2 ± 9.6 | 0.49 |
| *Blood biochemistry* | | | | |
| HbA$_{1c}$ (mmol/mol) | 34.8 ± 4.4 | 37.0 ± 4.5 | 32.9 ± 3.4 | 0.04 |
| Fasting plasma glucose, FPG (mmol/L) | 5.5 ± 0.8 | 5.7 ± 0.9 | 5.3 ± 0.6 | 0.20 |
| Fasting plasma insulin (µU/ml) | 10.5 ± 5.9 | 12.8 ± 5.5 | 8.4 ± 5.6 | 0.11 |
| HOMA2-IR | 1.4 ± 0.8 | 1.7 ± 0.8 | 1.1 ± 0.7 | 0.09 |

Results are mean ± SD. Level of significance set at p < 0.05. *Nonstandard abbreviations*: DXA, dual energy X-ray absorptiometry; MRI/S, magnetic resonance imaging and spectroscopy.

all women. Whilst fasting glucose:insulin ratio (FGIR), an index of IS, was not significantly correlated with PPF, fasting C-peptide:glucose ratio showed a borderline significant positive association (p = 0.07) with PPF (Fig 2E). A cleavage product of pro-insulin, C-peptide was a less variable marker of insulin secretion in the fasted state.

## Time course variations in blood biomarkers of pancreatic β-cell function and insulin secretion

During the two consecutive hyperglycaemic 30-min square-wave steps, glucose concentration increased from mean 5.5 ± 0.8 mmol/L in all women to the desired plateaux of 2.8 and 5.6 mmol/L above baseline and clamped for the following 30-min of the test (Fig 3A and 3E). Circulating plasma insulin and C-peptide concentrations (Fig 3B, 3C, 3F and 3G) followed a parallel pattern over the period of the test, with distinct peaks and a gradual plateau in hormone levels following each hyperglycaemic step. In comparison to circulating insulin and C-peptide respectively, ISR showed more rapid time-peak dynamics (Fig 3D and 3H). On average, the peak in the first-phase ISR was reached between 2–6 min after the initial glucose bolus and rapidly decreased thereafter to the pre-bolus value until the second 30-min step.

**Table 2. Summary characteristics of baseline anthropometry and body composition in women identified with MRI-determined %pancreatic fat above and below 4.5%.**

| | All (N = 19) | ≥ 4.5% (N = 9) | < 4.5% (N = 10) | *p value* |
|---|---|---|---|---|
| Age (y) | 47.3 ± 12.2 | 48.6 ± 9.1 | 46.1 ± 14.9 | 0.67 |
| *Anthropometry* | | | | |
| Bodyweight (kg) | 72.5 ± 13.6 | 74.6 ± 79.2 | 70.5 ± 16.9 | 0.53 |
| Height (m) | 1.64 ± 0.08 | 1.65 ± 0.07 | 1.62 ± 0.10 | 0.53 |
| Body-mass index, BMI (kg/m$^2$) | 27.0 ± 3.7 | 27.5 ± 3.2 | 26.6 ± 4.2 | 0.59 |
| Waist circumference (cm) | 88.6 ± 11.1 | 90.6 ± 9.6 | 86.8 ± 12.6 | 0.48 |
| Hip circumference (cm) | 100.7 ± 10.0 | 103.8 ± 7.5 | 97.9 ± 10.4 | 0.21 |
| Systolic blood pressure, SBP (mmHg) | 124 ± 24 | 128 ± 28 | 120 ± 20 | 0.46 |
| Diastolic blood pressure, DBP (mmHg) | 67 ± 14 | 65 ± 15 | 66 ± 13 | 0.91 |
| *Body composition—DXA* | | | | |
| Total body fat, DXA-TBF (kg) | 28.5 ± 9.2 | 30.8 ± 6.2 | 26.4 ± 11.2 | 0.31 |
| Total body fat, DXA-TBF (%) | 40.0 ± 6.2 | 42.5 ± 4.3 | 37.8 ± 7.0 | 0.10 |
| Abdominal adipose tissue, DXA-AAT (kg) | 2.4 ± 0.9 | 2.7 ± 0.6 | 2.2 ± 1.1 | 0.28 |
| Abdominal adipose tissue, DXA-AAT (%) | 45.0 ± 8.2 | 48.2 ± 5.2 | 42.2 ± 9.6 | 0.11 |
| *Body composition–MRI/S* | | | | |
| Visceral adipose tissue, MR-VAT (cm$^2$) | 69.5 ± 24.1 | 80.8 ± 20.1 | 59.4 ± 23.7 | 0.05 |
| Subcutaneous adipose tissue, MR-SAT (cm$^2$) | 165.8 ± 61.5 | 165.3 ± 52.3 | 166.3 ± 71.6 | 0.97 |
| Liver fat (%) | 7.7 ± 7.3 | 10.7 ± 8.6 | 5.0 ± 4.9 | 0.09 |
| Pancreatic fat % | 4.2 ± 1.7 | 5.8 ± 0.8 | 2.8 ± 0.7 | <0.0001 |
| Pancreas volume cm$^3$ | 74.3 ± 19.4 | 65.9 ± 22.9 | 81.8 ± 12.3 | 0.07 |
| Pancreas volume index, PVI | 42.0 ± 11.2 | 36.1 ± 11.5 | 47.3 ± 8.2 | 0.03 |
| *Blood biochemistry* | | | | |
| HbA$_{1c}$ (mmol/mol) | 34.8 ± 4.4 | 34.6 ± 2.9 | 35.1 ± 5.5 | 0.80 |
| Fasting plasma glucose, FPG (mmol/L) | 5.5 ± 0.8 | 5.6 ± 0.9 | 5.4 ± 0.6 | 0.54 |
| Fasting plasma insulin (μU/ml) | 10.5 ± 5.9 | 10.0 ± 4.1 | 10.9 ± 7.4 | 0.77 |
| HOMA2-IR | 1.4 ± 0.8 | 1.3 ± 0.5 | 1.4 ± 1.0 | 0.80 |

Results are mean ± SD. Level of significance set at p < 0.05. Group with ≥ 4.5% pancreatic fat includes n = 5 Chinese and n = 4 Caucasian women; group with < 4.5% pancreatic fat includes n = 4 Chinese and n = 6 Caucasian women. *Nonstandard abbreviations*: DXA, dual-energy X-ray absorptiometry; MRI/S, magnetic resonance imaging and spectroscopy.

Repeated measures linear mixed model analyses, adjusted for ethnicity and baseline concentrations, showed that whilst there was a significant change from baseline in plasma glucose (time, p < 0.0001, Fig 3I) and insulin (time, p < 0.0001, Fig 3J) in both low and high PPF groups, there was no significant difference between the two groups at any given time (overall interaction, PPF x time, both p > 0.05).

## First- and second-phase insulin secretion

The change from baseline (Δ) peak first-phase ISR, following the first hyperglycaemic episode (0.03 ± 0.03 nmol min$^{-1}$ m$^{-2}$), was significantly greater (p = 0.01) than the Δ peak second-phase ISR, from 30-min in response to an identical glucose bolus/gradient (0.01 ± 0.03 nmol min$^{-1}$ m$^{-2}$) (Fig 4A). This pattern of ISR follows that of plasma glucose, and corresponds to insulin and C-peptide levels (Fig 3B and 3C) secreted to counter the rise in glucose levels during the second step. There were no significant differences observed in the Δ peak first and second phases between the PPF groups (Fig 4B and 4C). Notably, the iAUC ISR for first- and second-phase insulin secretion phase was 0.04 units higher in the low compared to the high

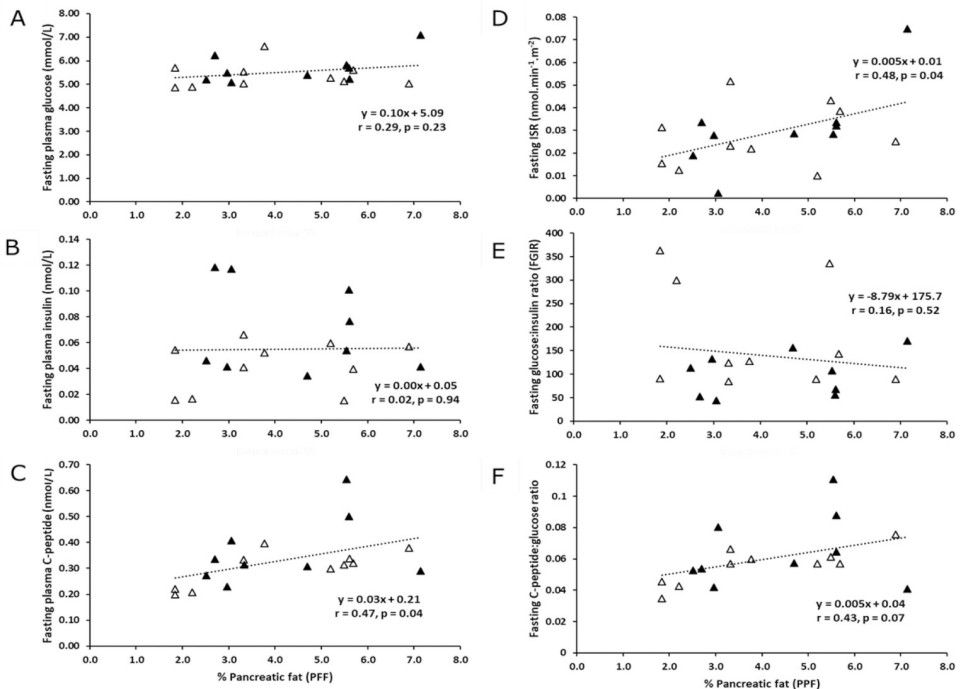

**Fig 2. Relationships between baseline circulating plasma glucoregulatory biomarkers, the rate of insulin secretion and indices of insulin sensitivity with MR-determined percentage pancreatic fat in all women (n = 10 Caucasian women open triangles; n = 9 Chinese women solid triangles).**

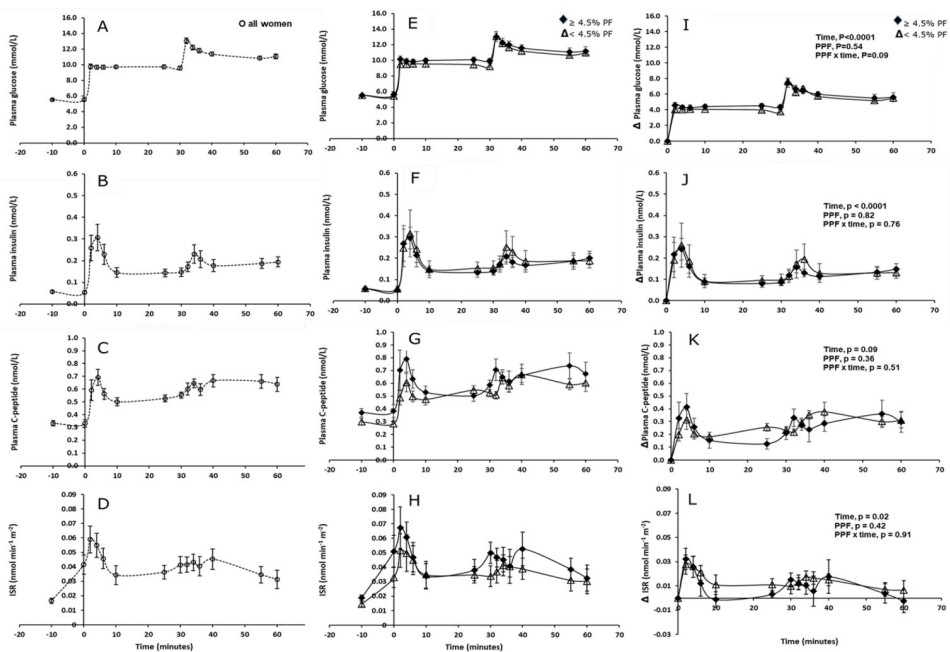

**Fig 3.** Time course (mean ± SEM) of blood biomarkers during each 30 minute square-wave hyperglycaemic step of the intravenous glucose tolerance test in all women (A-D—open circle, dash lines) in defined low and high percentage pancreatic fat (PPF) groups (E-H) and as change from baseline (I-L). MR-determined PPF groups are as follows: E-L; ≥ 4.5% pancreatic fat–solid diamond, solid line, < 4.5% pancreatic fat–open triangle, solid line.

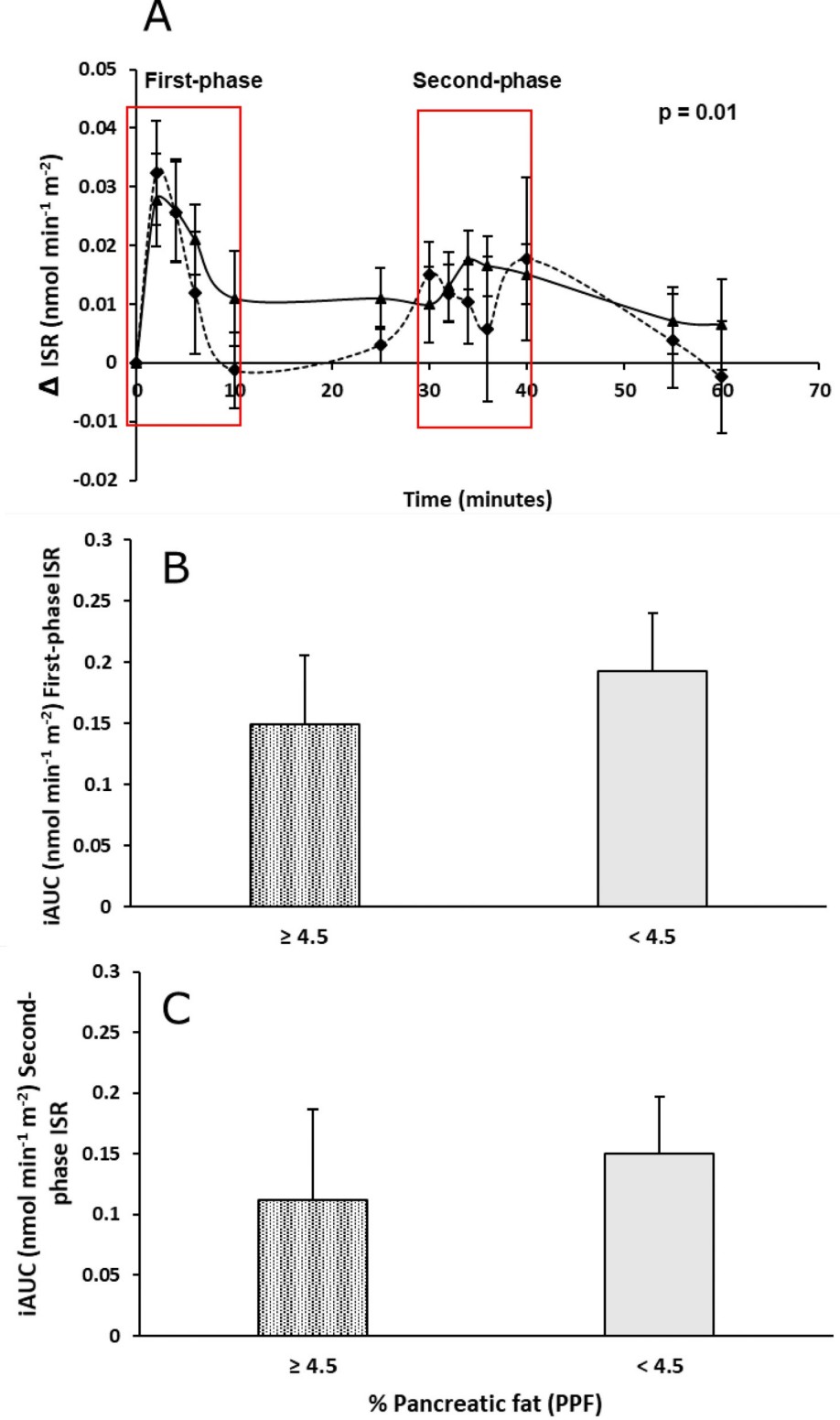

**Fig 4. Rate of insulin secretion over the 60-min test period in women according to MR-determined percentage pancreatic fat (PPF) cut off ($\geq$ 4.5% pancreatic fat–solid diamond, dash lines; < 4.5% pancreatic fat–solid triangle, solid line, A) and incremental area under the curves (iAUC) for first- and second-phase insulin response determined for each pancreatic fat sub-group (B, C).** The change from baseline ($\Delta$) peak first-phase ISR (0.03 $\pm$ 0.03 nmol min$^{-1}$ m$^{-2}$) was significantly greater (p = 0.01) than the $\Delta$ peak second-phase ISR (0.01 $\pm$ 0.03 nmol min$^{-1}$ m$^{-2}$).

PPF sub-group in both phases. Although of clinical relevance, the differences were not statistically significant (p = 0.58, iAUC first-phase; p = 0.67, iAUC second-phase).

## Discussion

To the best of our knowledge, this is the first study utilising the robust MR-opsy technique and defined high and low PPF groups to identify pancreatic β-cell response to two consecutive square-wave steps of hyperglycaemia. In this well-matched mixed ethnic cohort of women, the majority of whom were overweight or obese but with no diagnosis of T2DM, we did not observe adverse effects of higher non-adipose intra-pancreatic fat accumulation on IS, β-cell compensation and secretory function during an ivGTT. However, we confirmed concomitantly greater MR-VAT deposition, and liver fat content, in women with ≥ 4.5% pancreatic fat, that aligned with conclusions from the recent position statement by Neeland *et al.* [1] that these sites are important contributors of metabolic risk. Further exploration of early changes in β-cell function and insulin secretion, via the observed 0.04 units lower first phase ISR, may better our understanding of the contribution of pancreatic fat to, and as a putative early marker of, increased T2DM risk.

Increased susceptibility to developing T2DM at lower BMI and younger age has been reported in Chinese [37–39] than Caucasians, with visceral adiposity and its downstream effects considered likely contributors of these ethnic differences [2, 3]. However, in this ivGTT investigation, the anthropometric and body composition parameters were similar between the ethnic groups except for abdominal MR-SAT, which expectedly, was higher in Caucasian than Chinese women, consistent with a previous report [39]. Due consideration was applied to statistical models which were adjusted for ethnicity to ensure any influences on parameters of insulin secretion [6] were controlled. Additionally fasted glucose concentrations, reflective of endogenous glucose production (EGP), were also adjusted for in statistical analyses. While none of the women in this investigation had confirmed diagnosis of T2DM, they were identified with normo- or dysglycaemia based on the American Diabetes Association defined fasting plasma glucose criterion. Physiologically, maintenance of circulating glucose homeostasis involves the opposing effects of glucagon and insulin, secreted by pancreatic α- and β-cells respectively, on target tissues. Given that the women attended the HNU, following an overnight fast, we expect that any hypoglycaemia over the period may have initiated a decrease in insulin secretion with a corresponding increase in glucagon to increase hepatic EGP via glycogenolysis and gluconeogenesis.

Measured circulating plasma insulin represents the balance between the ISR with hepatic and extra-hepatic clearance, with hepatic clearance known to undergo change following the stimulation of endogenous insulin secretion [40]. In contrast C-peptide, which is co-secreted in equimolar concentration with insulin into the portal vein, avoids hepatic degradation and is entirely cleared at a relatively constant rate in peripheral tissues [41]. Hence, peripheral plasma C-peptide concentrations have been used to estimate the rate at which insulin is secreted and to measure β-cell function. C-peptide deconvolution is widely used in the estimation of prehepatic (total) insulin secretion and is based on the premise that C-peptide is secreted in a central compartment from which it distributes into a peripheral extravascular compartment with lower variability [36]. Baseline C-peptide measured in the study was between 0.3–0.6 nmol/L, similar to that previously reported in healthy individuals [42], and levels were maintained below 1.0 nmol/L over the testing period. Baseline C-peptide concentration as well as ISR were found to be significantly and positively associated with the PPF. The positive association might be considered unexpected given that increased PPF has often been related to altered β-cell function. Nonetheless, the relationships are physiologically relevant particularly IS from the C-

peptide:glucose ratio. During insulin resistant states, increased β-cell response to (endogenous) glucose and decreased insulin clearance would be adaptive mechanisms for the maintenance of normal glucose tolerance.

Following glucose stimulation, insulin is released in a biphasic manner from pancreatic β-cells [43] that is considered to reflect two populations of insulin secretory granules; these are the readily releasable pool which is responsible for the initial (first-phase) of insulin secretion and a second reserve pool which accounts for the more prolonged (second-phase) of insulin secretion [44]. The reduction in first-phase insulin release is considered as the earliest detectable defect in β-cell secretory response [45] and thought to further impact postprandial plasma glucose excursions [46]. Whilst previous studies using OGTT [4–6] have reported negative correlations between pancreatic fat and insulin secretion, no studies have yet documented the dynamic relationship between pancreatic fat and early first-phase ISR as done in this study. Furthermore, the overall temporal ISR dynamics during the stepped ivGTT, with a greater first-phase to second-phase insulin response, is in alignment with that previously reported in healthy volunteers [33]. While, increased first-phase insulin response has been demonstrated to accompany decreased pancreatic fat over 8-weeks, this was observed in individuals who were already diagnosed with frank T2DM [47] and not the earlier stages of dysglycaemia.

There is an emerging role of pancreatic fat in the aetiology of T2DM [14, 47]. A recent meta-analysis [16] in 1209 healthy individuals, albeit from studies using different modalities for assessing pancreatic fat (including computed tomography, endoscopic ultrasound and MRI), reported the weighted mean as 4.48 ± 0.87% for healthy pancreata. On the basis of this, we utilised a 4.5% cut-off for internal comparison within our cohort.

Limitations of this study include the small, single gender cohort which may not allow generalization of the data and findings. At present, MR methodologies do not allow discrimination between intracellular fat accumulation in β-cells and adipose tissue infiltration, with histology the gold standard. Nonetheless, the MR-opsy technique utilised in this study for determination of pancreatic fat is robust and reproducible, along side the ivGTT which similarly is highly reproducible for assessing β-cell sensitivity to glucose. We acknowledge that individuals with MR-quantified pancreatic fat and frank T2DM may have a differential insulin secretory response, than that observed in our study, which may provide other potential mechanisms of interest and warrants further investigation. The use of C-peptide kinetics may at times not directly correlate with circulating levels of biologically active insulin [48]. It may underestimate the rate of insulin secretion particularly in conditions where insulin release is rapidly changing, for example immediately following an intravenous glucose bolus. On the other hand, it may overestimate insulin secretion under conditions where insulin release is rapidly declining.

In conclusion, this investigation provides an important insight into the temporal dynamics of first- and second-phase insulin secretion, in the presence of pancreatic fat. Whilst we show no adverse effects of pancreatic fat on β-cell function in this cohort of women who were overweight or obese but did not have T2DM, we consider that there may be some salient effects on first-phase insulin response in early stages of dysglycaemia that warrant further investigation.

## Acknowledgments

We are most grateful to the volunteers that participated in this study. We sincerely thank Prof Jun Lu for developing the MRI/S protocol for pancreas image acquisition in this study; Research Nurses Madhavi Bolleneni and Clarence Vivar for their assistance during the intravenous glucose tolerance tests; The Liggins Institute Laboratory, University of Auckland, for biochemical analyses.

## Author Contributions

**Conceptualization:** Carl N. Peters, Sally D. Poppitt.

**Data curation:** Ivana R. Sequeira, Wilson Yip, Kieren G. Hollingsworth, Sally D. Poppitt.

**Formal analysis:** Ivana R. Sequeira, Yannan Jiang.

**Funding acquisition:** Sally D. Poppitt.

**Investigation:** Ivana R. Sequeira, Wilson Yip, Louise W. Lu.

**Methodology:** Ivana R. Sequeira, Carl N. Peters, Benjamin S. Aribsala, Kieren G. Hollingsworth.

**Project administration:** Garth J. S. Cooper, Sally D. Poppitt.

**Software:** Benjamin S. Aribsala, Kieren G. Hollingsworth.

**Supervision:** Ivana R. Sequeira, Lindsay D. Plank, Carl N. Peters, Kieren G. Hollingsworth, Sally D. Poppitt.

**Validation:** Ivana R. Sequeira, Wilson Yip.

**Writing – original draft:** Ivana R. Sequeira.

**Writing – review & editing:** Ivana R. Sequeira, Rinki Murphy, Lindsay D. Plank, Garth J. S. Cooper, Kieren G. Hollingsworth, Sally D. Poppitt.

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
