## [Decision Letter · Decision Letter 0]

17 Jun 2022

PONE-D-21-36957Exploring the relationship between pancreatic fat and insulin secretion in overweight or obese women who did not have type 2 diabetesPLOS ONE

Dear Dr. Sequeira,

Thank you for submitting your manuscript to PLOS ONE. After careful consideration, we feel that it has merit but does not fully meet PLOS ONE’s publication criteria as it currently stands. Therefore, we invite you to submit a revised version of the manuscript that addresses the points raised during the review process.

 Please see the comments from one reviewer below. Please note that their final comment about blurry figures does not need to be addressed - the blurriness in the reviewer PDF is an artefact of the PDF generation, and the original images are of high enough quality. Please note that we have only been able to secure a single reviewer to assess your manuscript. We are issuing a decision on your manuscript at this point to prevent further delays in the evaluation of your manuscript. Please be aware that the editor who handles your revised manuscript might find it necessary to invite additional reviewers to assess this work once the revised manuscript is submitted. However, we will aim to proceed on the basis of this single review if possible.

We look forward to receiving your revised manuscript.

Kind regards,

Hanna Landenmark

Staff Editor

PLOS ONE

Journal Requirements:

Additional Editor Comments:

Please note that outmoded terms and potentially stigmatizing labels should be changed to more current, acceptable terminology. Examples: “Caucasian” should be changed to “white” or “of [Western] European descent” (as appropriate); “cancer victims” should be changed to “patients with cancer.” (https://journals.plos.org/plosone/s/submission-guidelines#loc-human-subjects-research)

Reviewers' comments:

Reviewer's Responses to Questions

**Comments to the Author**

1. Is the manuscript technically sound, and do the data support the conclusions?

Reviewer #1: Partly

2. Has the statistical analysis been performed appropriately and rigorously? 

Reviewer #1: I Don't Know

3. Have the authors made all data underlying the findings in their manuscript fully available?

Reviewer #1: Yes

4. Is the manuscript presented in an intelligible fashion and written in standard English?

Reviewer #1: Yes

5. Review Comments to the Author

Reviewer #1: In this study, the authors aimed to determine whether pancreatic fat negatively affects beta cell function in women with overweight or obesity but without T2D. Twenty women with NGT or IFG were included in the study. Low and high pancreatic fat was assessed by MRI and insulin secretion was assessed by ivGTT.

Use of ivGTT and MRI to assess insulin secretion and pancreas fat is an advantage of the study. However, there are several major critiques.

1. The study design was not clearly described. If this is a subpopulation analysis of the original study (TOFI Asia cross-sectional study), brief description of the original study should be included in the methods section. The sample size calculation should also be described clearly.

2. The small sample size is a major limitation. In addition, the inclusion of different glucose tolerance status (NGT and IFG) and different ethnic groups made interpretation of data more complex. It is difficult to conclude based on this small sample size and the authors are highly recommended to increase the sample size to lead more solid conclusion. At least, the term “preliminary results” or “pilot study” should be included in the title.

3. As the incremental AUCs for both first and second phase ISR were not significantly different between the high and low PPF groups, their conclusion “early signs of decreased biphasic insulin secretion in those with pancreatic fat >=4.5% may be of clinical importance” was not supported by the data and should be removed.

4. Uploaded figures are blur and difficult to see. More clear images should be provided.

6. PLOS authors have the option to publish the peer review history of their article (what does this mean?). If published, this will include your full peer review and any attached files.

Reviewer #1: No

---

## [Author Response · Author response to Decision Letter 0]

8 Aug 2022

We would like to express our gratitude to the reviewer for providing us with insightful comments, and their efforts towards improving our manuscript. We have addressed and provided detailed responses to the feedback received below in the highlighted clean and track change versions of the revised manuscript. Please note that all pagination and line numbers refer to the submitted clean highlighted revised version. 

Comments to the Author

In this study, the authors aimed to determine whether pancreatic fat negatively affects beta cell function in women with overweight or obesity but without T2D. Twenty women with NGT or IFG were included in the study. Low and high pancreatic fat was assessed by MRI and insulin secretion was assessed by ivGTT. Use of ivGTT and MRI to assess insulin secretion and pancreas fat is an advantage of the study. However, there are several major critiques.

1. The study design was not clearly described. If this is a subpopulation analysis of the original study (TOFI Asia cross-sectional study), brief description of the original study should be included in the methods section. The sample size calculation should also be described clearly.

AUTHOR RESPONSE: We note the comment by the reviewer and agree that this information was not clearly presented. We have now briefly expanded on this in the MATERIALS AND METHODS Study design and protocol section on Page 9, Line 156–164 “All women enrolled in the TOFI_Asia MR cohort attended the HNU clinic following an overnight fast; body weight, height, waist and hip circumferences, and blood pressure (BP) were recorded, and a fasted blood sample for determination of HbA1c collected, stored at -80°C pending batch analysis on completion of the study using capillary electrophoresis (Cap2FP, IDF, France). Total and abdominal adiposity were assessed using dual-energy X-ray absorptiometry (DXA) (iDXA, GE Healthcare, WI, USA) at the Body Composition Laboratory, University of Auckland. Detailed abdominal MRI/S scans were conducted in the fasted state, within 1 week of the clinic visit, on a 3T Magnetom Skyra scanner, VE 11A (Siemens, BY, Germany) at the Centre for Advanced MRI (CAMRI), University of Auckland..” 

And further on Page 11, Line 205-222 “Following MR scanning, women who provided additional written informed consent were scheduled for their ivGTT within 7 days. A stepped insulin secretion test was performed as previously described (14, 33). In fasted participants, two 18G venous cannulae were inserted into the antecubital vein for infusion and the contralateral wrist vein for arterialised blood sampling. The hand was kept warm during the entire test to maintain arterialisation of venous blood. Two consecutive, 30-min square-wave steps of hyperglycaemia (2.8 and 5.6 mmol/L above baseline) were achieved using a bolus dose of 25% dextrose (Baxter Healthcare, NSW, Australia). The priming bolus dose for each step was calculated as 0.378 g/kg body weight of 25% dextrose and administered over one min to avoid thrombophlebitis. Post-administration of the bolus dose, plasma glucose concentrations were maintained at the desired plateau with a variable 25% dextrose infusion (Carefusion, Alaris®, BBRK, UK). Blood samples for determination of plasma glucose, insulin and C-peptide were obtained every 2 min for the first 10 min and every 5 min over the next 20 min of each step. Two Caucasian women did not successfully complete the second 30-min of the test, due to the inability to obtain arterialised venous blood. We were unable to obtain sufficient blood samples for analysis and estimation of insulin secretion rate (ISR) from one Caucasian woman, who was therefore excluded from the statistical analysis. Therefore, final data presented are for 19 women (9 Chinese and 10 Caucasian) (Fig. 1).”

Whilst there was no a priori sample size calculation conducted for the study, we performed an informal assessment of sample size based on a similar prior study published by members of our research team which also investigated beta cell function in a group of 11 individuals: Lim, Hollingsworth, Aribisala, et al., Reversal of type 2 diabetes: normalisation of beta cell function in association with decreased pancreas and liver triacylglycerol. Diabetologia 2011, doi: 10.1007/s00125-011-2204-7. The informal sample size assessment was based on first-phase insulin response at baseline (mean: 0.19 ± 0.02 SEM nmol min−1 m−2) following dietary intervention (mean: 0.46 ± 0.07 SEM nmol min−1 m−2). Based on the Lim data, to detect effect size 0.27 nmol min−1 m−2 as significant would require a minimum of n = 10 individuals. Modelling based on a more conservative effect size of ~50% of the Lim data (0.14 nmol min−1 m−2 ) and variance at pre-intervention baseline (0.07 nmol min−1 m−2) generated a required sample size of n = 14 for 2 independent groups, and effect size of ~25% of Lim generated a required sample size of n = 23 per group. Based on these estimates we continued to recruit individuals from the TOFI_Asia cohort until we had enrolled 22 individuals. 

Hence, all women from the TOFI_Asia MR study who underwent an MRI scan and provided informed consent to undertake a 1 hr-ivGTT were included in the cohort. We have now clarified this in the MATERIALS AND METHODS Study Population section on Page 8, line 140 – 153 “Of the 202 women who were enrolled in the TOFI_Asia cross-sectional study, a subset of 68 women (34 Chinese and 34 Caucasian) successfully joined the TOFI_Asia MR cohort and underwent abdominal MRI scans for assessment of pancreatic fat. Of these, 22 women (9 Chinese and 13 Caucasian) provided written informed consent and were enrolled into the current ivGTT cohort, for determination of pancreatic β-cell function as a measurement of insulin secretion (Fig. 1). Informal power calculations showed a minimum of 10 individuals would be required to detect a physiologically significant increase in first phase insulin response. Inclusion criteria was based on the TOFI_Asia MR study (26) and as such the ivGTT cohort were 20-70 years of age, body mass index (BMI) 20–35 kg/m2, normoglycaemic (fasting plasma glucose < 5.6 mmol/L) or impaired fasting glucose (IFG; 5.6 – 6.9 mmol/L) (27) and MRI-assessed percentage pancreatic fat (PPF) classified as < 4.5 % or ≥ 4.5 %. PPF could not be analysed from scans obtained from 2 women due to motion related artefact, and hence 20 women (9 Chinese and 11 Caucasian) had an assessment of pancreatic β-cell function using ivGTT and were included in the current analyses (Fig. 1).”

2. The small sample size is a major limitation. In addition, the inclusion of different glucose tolerance status (NGT and IFG) and different ethnic groups made interpretation of data more complex. It is difficult to conclude based on this small sample size and the authors are highly recommended to increase the sample size to lead more solid conclusion. At least, the term “preliminary results” or “pilot study” should be included in the title.

AUTHOR RESPONSE: We note the comment made by the reviewer and are in agreement that sample size is indeed a limitation of the study and have acknowledged this on Page 18, Line 385-386 “Limitations of this study include the small, single gender cohort which may not allow generalization of the data and findings.” We have taken on board the reviewers suggestion and have now revised the title of the paper on Page 1, Line 1-3 “Exploring the relationship between pancreatic fat and insulin secretion in overweight or obese women without type 2 diabetes: a preliminary investigation of the TOFI_Asia cohort”

We also note the reviewer’s suggestion to increase the sample size. Our study was completed in 2017 and although presently unable to add to the present cohort, we are undertaking 3-yr follow-up assessments on the cohort which we are looking to publish in early 2023. 

Additionally, although we have a small sample size of 20 women (9 Chinese and 13 Caucasian), a strength of our study is that we undertook statistical analyses specifically using repeated measures linear mixed models, that were adjusted for ethnicity and the average fasted glucose concentration (reflective of endogenous glucose production), in women identified with PPF above and below our defined high/low PPF cut-off. This has been detailed in the Statistical Analysis section Page 12, line 242-246, “Changes from baseline (Δ) in glucose, insulin, C-peptide and ISR over the 60-min test were evaluated using repeated measures linear mixed models, adjusted for ethnicity and the average baseline (-10 min and 0 min) concentration, in women identified with PPF above and below our defined high/low PPF cut-off. The interaction effect between the two PPF groups and time points was tested in the model.”

3. As the incremental AUCs for both first and second phase ISR were not significantly different between the high and low PPF groups, their conclusion “early signs of decreased biphasic insulin secretion in those with pancreatic fat >=4.5% may be of clinical importance” was not supported by the data and should be removed.

AUTHOR RESPONSE: We have taken on board the reviewer’s feedback and agree that the conclusion unsupported and requires to be revised in the Conclusion of the Abstract. The sentence has been re-phrased on Page 4, Line 64-67 “In women with overweight or obesity but without type 2 diabetes, PPF did not modify β-cell function as determined by ivGTT-assessed ISR. However, the salient feature in biphasic insulin secretion in those with ≥4.5% PPF may be of clinical importance, particularly in early stages of dysglycaemia may warrant further investigation.” 

4. Uploaded figures are blur and difficult to see. More clear images should be provided.

AUTHOR RESPONSE: We thank the reviewer for the attention to detail. We would like to intimate the reviewer that upon receipt of the reviewer comments we were informed by the Staff Editor (Hanna Landenmark) that the blurriness in the reviewer PDF was an artefact of the PDF generation. They confirmed that the original images that we uploaded onto the PLOS One system are of high enough quality for the Journal and hence informed us that we did not need to revise the images further.

---

## [Editor Report · Decision Letter 1]

15 Aug 2022

PONE-D-21-36957R1

Exploring the relationship between pancreatic fat and insulin secretion in overweight or obese women without type 2 diabetes: a preliminary investigation of the TOFI_Asia cohort

PLOS ONE

Dear Dr. Sequeira,

Thank you for submitting your manuscript to PLOS ONE. After careful consideration, we have decided that your manuscript does not meet our criteria for publication and must therefore be rejected.

I am sorry that we cannot be more positive on this occasion, but hope that you appreciate the reasons for this decision.

Kind regards,

Yoshifumi Saisho

Guest Editor

PLOS ONE

Additional Editor Comments:

The authors have responded to the comments from the reviewer partly. However, it is well-known that there are significant differences in insulin sensitivity and insulin secretion ability between the Chinese and Caucasian. Therefore, the findings of this study may be flawed and not scientifically robust.
---

## [Author Response · Author response to Decision Letter 1]

30 Aug 2022

Reviewers' comments:

We would like to express our gratitude to the reviewer for providing us with insightful comments, and their efforts towards improving our manuscript. We have addressed and provided detailed responses to the feedback received below in the highlighted clean and track change versions of the revised manuscript. Please note that all pagination and line numbers refer to the submitted clean highlighted revised version. 

Comments to the Author

In this study, the authors aimed to determine whether pancreatic fat negatively affects beta cell function in women with overweight or obesity but without T2D. Twenty women with NGT or IFG were included in the study. Low and high pancreatic fat was assessed by MRI and insulin secretion was assessed by ivGTT. Use of ivGTT and MRI to assess insulin secretion and pancreas fat is an advantage of the study. However, there are several major critiques.

1. The study design was not clearly described. If this is a subpopulation analysis of the original study (TOFI Asia cross-sectional study), brief description of the original study should be included in the methods section. The sample size calculation should also be described clearly.

AUTHOR RESPONSE: We note the comment by the reviewer and agree that this information was not clearly presented. We have now briefly expanded on this in the MATERIALS AND METHODS Study design and protocol section on Page 9, Line 156–164 “All women enrolled in the TOFI_Asia MR cohort attended the HNU clinic following an overnight fast; body weight, height, waist and hip circumferences, and blood pressure (BP) were recorded, and a fasted blood sample for determination of HbA1c collected, stored at -80°C pending batch analysis on completion of the study using capillary electrophoresis (Cap2FP, IDF, France). Total and abdominal adiposity were assessed using dual-energy X-ray absorptiometry (DXA) (iDXA, GE Healthcare, WI, USA) at the Body Composition Laboratory, University of Auckland. Detailed abdominal MRI/S scans were conducted in the fasted state, within 1 week of the clinic visit, on a 3T Magnetom Skyra scanner, VE 11A (Siemens, BY, Germany) at the Centre for Advanced MRI (CAMRI), University of Auckland..” 

And further on Page 11, Line 205-222 “Following MR scanning, women who provided additional written informed consent were scheduled for their ivGTT within 7 days. A stepped insulin secretion test was performed as previously described (14, 33). In fasted participants, two 18G venous cannulae were inserted into the antecubital vein for infusion and the contralateral wrist vein for arterialised blood sampling. The hand was kept warm during the entire test to maintain arterialisation of venous blood. Two consecutive, 30-min square-wave steps of hyperglycaemia (2.8 and 5.6 mmol/L above baseline) were achieved using a bolus dose of 25% dextrose (Baxter Healthcare, NSW, Australia). The priming bolus dose for each step was calculated as 0.378 g/kg body weight of 25% dextrose and administered over one min to avoid thrombophlebitis. Post-administration of the bolus dose, plasma glucose concentrations were maintained at the desired plateau with a variable 25% dextrose infusion (Carefusion, Alaris®, BBRK, UK). Blood samples for determination of plasma glucose, insulin and C-peptide were obtained every 2 min for the first 10 min and every 5 min over the next 20 min of each step. Two Caucasian women did not successfully complete the second 30-min of the test, due to the inability to obtain arterialised venous blood. We were unable to obtain sufficient blood samples for analysis and estimation of insulin secretion rate (ISR) from one Caucasian woman, who was therefore excluded from the statistical analysis. Therefore, final data presented are for 19 women (9 Chinese and 10 Caucasian) (Fig. 1).”

Whilst there was no a priori sample size calculation conducted for the study, we performed an informal assessment of sample size based on a similar prior study published by members of our research team which also investigated beta cell function in a group of 11 individuals: Lim, Hollingsworth, Aribisala, et al., Reversal of type 2 diabetes: normalisation of beta cell function in association with decreased pancreas and liver triacylglycerol. Diabetologia 2011, doi: 10.1007/s00125-011-2204-7. The informal sample size assessment was based on first-phase insulin response at baseline (mean: 0.19 ± 0.02 SEM nmol min−1 m−2) following dietary intervention (mean: 0.46 ± 0.07 SEM nmol min−1 m−2). Based on the Lim data, to detect effect size 0.27 nmol min−1 m−2 as significant would require a minimum of n = 10 individuals. Modelling based on a more conservative effect size of ~50% of the Lim data (0.14 nmol min−1 m−2 ) and variance at pre-intervention baseline (0.07 nmol min−1 m−2) generated a required sample size of n = 14 for 2 independent groups, and effect size of ~25% of Lim generated a required sample size of n = 23 per group. Based on these estimates we continued to recruit individuals from the TOFI_Asia cohort until we had enrolled 22 individuals. 

Hence, all women from the TOFI_Asia MR study who underwent an MRI scan and provided informed consent to undertake a 1 hr-ivGTT were included in the cohort. We have now clarified this in the MATERIALS AND METHODS Study Population section on Page 8, line 140 – 153 “Of the 202 women who were enrolled in the TOFI_Asia cross-sectional study, a subset of 68 women (34 Chinese and 34 Caucasian) successfully joined the TOFI_Asia MR cohort and underwent abdominal MRI scans for assessment of pancreatic fat. Of these, 22 women (9 Chinese and 13 Caucasian) provided written informed consent and were enrolled into the current ivGTT cohort, for determination of pancreatic β-cell function as a measurement of insulin secretion (Fig. 1). Informal power calculations showed a minimum of 10 individuals would be required to detect a physiologically significant increase in first phase insulin response. Inclusion criteria was based on the TOFI_Asia MR study (26) and as such the ivGTT cohort were 20-70 years of age, body mass index (BMI) 20–35 kg/m2, normoglycaemic (fasting plasma glucose < 5.6 mmol/L) or impaired fasting glucose (IFG; 5.6 – 6.9 mmol/L) (27) and MRI-assessed percentage pancreatic fat (PPF) classified as < 4.5 % or ≥ 4.5 %. PPF could not be analysed from scans obtained from 2 women due to motion related artefact, and hence 20 women (9 Chinese and 11 Caucasian) had an assessment of pancreatic β-cell function using ivGTT and were included in the current analyses (Fig. 1).”

2. The small sample size is a major limitation. In addition, the inclusion of different glucose tolerance status (NGT and IFG) and different ethnic groups made interpretation of data more complex. It is difficult to conclude based on this small sample size and the authors are highly recommended to increase the sample size to lead more solid conclusion. At least, the term “preliminary results” or “pilot study” should be included in the title.

AUTHOR RESPONSE: We note the comment made by the reviewer and are in agreement that sample size is indeed a limitation of the study and have acknowledged this on Page 18, Line 385-386 “Limitations of this study include the small, single gender cohort which may not allow generalization of the data and findings.” We have taken on board the reviewers suggestion and have now revised the title of the paper on Page 1, Line 1-3 “Exploring the relationship between pancreatic fat and insulin secretion in overweight or obese women without type 2 diabetes: a preliminary investigation of the TOFI_Asia cohort”

We also note the reviewer’s suggestion to increase the sample size. Our study was completed in 2017 and although presently unable to add to the present cohort, we are undertaking 3-yr follow-up assessments on the cohort which we are looking to publish in early 2023. 

Additionally, although we have a small sample size of 20 women (9 Chinese and 13 Caucasian), a strength of our study is that we undertook statistical analyses specifically using repeated measures linear mixed models, that were adjusted for ethnicity and the average fasted glucose concentration (reflective of endogenous glucose production), in women identified with PPF above and below our defined high/low PPF cut-off. This has been detailed in the Statistical Analysis section Page 12, line 242-246, “Changes from baseline (Δ) in glucose, insulin, C-peptide and ISR over the 60-min test were evaluated using repeated measures linear mixed models, adjusted for ethnicity and the average baseline (-10 min and 0 min) concentration, in women identified with PPF above and below our defined high/low PPF cut-off. The interaction effect between the two PPF groups and time points was tested in the model.”

3. As the incremental AUCs for both first and second phase ISR were not significantly different between the high and low PPF groups, their conclusion “early signs of decreased biphasic insulin secretion in those with pancreatic fat >=4.5% may be of clinical importance” was not supported by the data and should be removed.

AUTHOR RESPONSE: We have taken on board the reviewer’s feedback and agree that the conclusion unsupported and requires to be revised in the Conclusion of the Abstract. The sentence has been re-phrased on Page 4, Line 64-67 “In women with overweight or obesity but without type 2 diabetes, PPF did not modify β-cell function as determined by ivGTT-assessed ISR. However, the salient feature in biphasic insulin secretion in those with ≥4.5% PPF may be of clinical importance, particularly in early stages of dysglycaemia may warrant further investigation.” 

4. Uploaded figures are blur and difficult to see. More clear images should be provided.

AUTHOR RESPONSE: We thank the reviewer for the attention to detail. We would like to intimate the reviewer that upon receipt of the reviewer comments we were informed by the Staff Editor (Hanna Landenmark) that the blurriness in the reviewer PDF was an artefact of the PDF generation. They confirmed that the original images that we uploaded onto the PLOS One system are of high enough quality for the Journal and hence informed us that we did not need to revise the images further. 

Editor Comments:

We, the authors, are very disappointed to hear that our manuscript “Exploring the relationship between pancreatic fat and insulin secretion in overweight or obese women without type 2 diabetes: a preliminary investigation of the TOFI_Asia cohort” has following revision [PONE-D-21-36957R1], not been considered suitable for publication in PlosOne without any opportunity to further comment and address concerns raised by the Guest Editor. This is also especially given that we submitted our manuscript to the Journal on 21st November 2021 and subsequently received an email 18th June 2022, seven months later, to inform us of the decision from Reviewer 1.

In accordance, the manuscript was improved based on the feedback received by Reviewer 1 with all detailed responses and revisions to comments provided. 

While we note difficulties the Journal may have encountered during the review process to procure reviewers, we would like to respectfully respond to the comment provided by the Guest Editor “The authors have responded to the comments from the reviewer partly. However, it is well-known that there are significant differences in insulin sensitivity and insulin secretion ability between the Chinese and Caucasian. Therefore, the findings of this study may be flawed and not scientifically robust” 

(i) Firstly, detailed responses to reviewer comments have been provided which mainly were around expanding on protocol and study design with inclusion of sample size calculation. The manuscript was duly revised to incorporate all of the feedback received with the following amendments made to the following sections:

- MATERIALS AND METHODS Study design and protocol section on Page 9, Line 156–164 and Page 11, Line 205-222

- MATERIALS AND METHODS Study Population section on Page 8, line 140 – 153

- Details of sample size calculation provided and described in MATERIALS AND METHODS Study Population section on Page 8, line 140 – 153 

- Limitation of the study have been acknowledged on Page 18, Line 385-386 

- Title of the manuscript revised on Page 1, Line 1-3

The Guest Editor’s response “The authors have responded to the comments from the reviewer partly.” The Editor does not provide any indication as to what ‘partly’ responds means. Specifically, to which question(s) does the Guest Editor refer to? This does not allow us to make any further revisions other than those stated in the detailed response to the reviewer comments. 

(ii) Secondly, while there are some publications that conclude there may be ethnic differences in insulin sensitivity and insulin secretion, the aims and objectives of our manuscript is not to decipher ethnic differences. Despite having enrolled women of two ethnicities in the study we conduct a preliminary investigation to explore the relationship between ectopic fat, particularly pancreatic fat and insulin secretion using the gold standard methods in women who are not diagnosed with by type 2 diabetes. 

This has been detailed in the original version of the manuscript in the following sections:

- INTRODUCTION on Page 7, Line 121-132.

- RESULTS on Page 13, Line 261 – 265 and 271 – 277. 

The results add further value and address existing gaps in our understanding of the role of pancreas fat and its adverse effects on EARLY stages of dysglycaemia which has not yet been previously reported using the current methods. 

This has been detailed in the original version of the manuscript in the following sections:

- RESULTS on Page 15, Line 299 – 303 and 306 – 315. 

And in the duly revised version of the Conclusion of the Abstract on Page 4, Line 64-67 following the reviewers comment “In women with overweight or obesity but without type 2 diabetes, PPF did not modify β-cell function as determined by ivGTT-assessed ISR. However, the salient feature in biphasic insulin secretion in those with ≥4.5% PPF may be of clinical importance, particularly in early stages of dysglycaemia may warrant further investigation.” 

The Editor does not seem to have not taken this into consideration based on the comment provided “ However, it is well-known that there are significant differences in insulin sensitivity and insulin secretion ability between the Chinese and Caucasian”. 

(iii) Thirdly, we have reported in our data that the cohort of women (Chinese and Caucasian) were matched across groups and ethnicity was factored into our statistical analysis. 

A strength of our study is that we undertook statistical analyses specifically using repeated measures linear mixed models, that were adjusted for ethnicity and the average fasted glucose concentration (reflective of endogenous glucose production), in women identified with pancreatic fat percentage above and below our defined high/low cut-off. 

- This has been detailed in the Statistical Analysis section Page 12, line 242-246.

We therefore do not consider that the Editors final comment “Therefore, the findings of this study may be flawed and not scientifically robust” is well justified. 

We would very much appreciate due consideration of the decision of our manuscript in light of the above response, the seven month period that our manuscript has been under consideration, as well as the improvements that we have instituted to further refine the manuscript. Please note that all page and line numbers listed above correspond to the clean untracked version of the submitted revised manuscript.

Thank you for your consideration of our manuscript and we look forward to your response.

---

## [Editor Report · Decision Letter 2]

1 Dec 2022

Exploring the relationship between pancreatic fat and insulin secretion in overweight or obese women without type 2 diabetes: a preliminary investigation of the TOFI_Asia cohort

PONE-D-21-36957R2

Dear Dr. Ivana Roosevelt Sequeira,

After careful re-evaluation, we’re pleased to inform you that your manuscript has been judged scientifically suitable for publication and will be formally accepted for publication once it meets all outstanding technical requirements. A minor change (add "mellitus" after "diabetes" throughout the paper) is required.

Kind regards,

Paolo Magni

Academic Editor

PLOS ONE

Additional Editor Comments (optional):

After careful re-evaluation I believe that the paper has been properly improved.

Minor change:

in the Title and in the Text, add "mellitus" after diabetes. The abbreviation should then read T2DM.
---

## [Editor Report · Acceptance letter]

22 Dec 2022

PONE-D-21-36957R2 

Exploring the relationship between pancreatic fat and insulin secretion in overweight or obese women without type 2 diabetes mellitus: a preliminary investigation of the TOFI_Asia cohort 

Dear Dr. Sequeira:

I'm pleased to inform you that your manuscript has been deemed suitable for publication in PLOS ONE. Congratulations! Your manuscript is now with our production department. 

Kind regards, 

on behalf of

Prof. Paolo Magni 

Academic Editor

PLOS ONE